# A Unified Probabilistic Framework for Volcanic Hazard and Eruption Forecasting

Warner Marzocchi[1], Jacopo Selva[2], Thomas H. Jordan[3]

[1] Department of Earth, Environmental, and Resources Sciences, University of Naples, Federico II, Complesso di Monte
Sant'Angelo, Via Cupa Nuova Cintia, 21 - 80126 Napoli, Italy
[2] Istituto Nazionale di Geofisica e Vulcanologia, Via Donato Creti 12, 40128 Bologna, Italy
[3] Department of Earth Sciences Southern California Earthquake Center, University of Southern California, Los Angeles,
California 90089

*Correspondence to*: Warner Marzocchi (warner.marzocchi@unina.it)

**Abstract.** The main purpose of this article is to emphasize the importance of clarifying the probabilistic framework adopted for volcanic hazard and eruption forecasting. Eruption forecasting and volcanic hazard analysis seek to quantify the deep uncertainties that pervade the modeling of pre-, sin- and post-eruptive processes. These uncertainties can be differentiated into three fundamental types: (1) the natural variability of volcanic systems, usually represented as stochastic processes with parameterized distributions (*aleatory variability*); (2) the uncertainty in our knowledge of how volcanic systems operate and evolve, often represented as subjective probabilities based on expert opinion (*epistemic uncertainty*); and (3) the possibility that our forecasts are wrong owing to behaviors of volcanic processes about which we are completely ignorant and, hence, cannot quantify in terms of probabilities (*ontological error*). Here we put forward a probabilistic framework for hazard analysis recently proposed by Marzocchi & Jordan (2014), which unifies the treatment of all three types of uncertainty. Within this framework, an eruption forecasting or a volcanic hazard model is said to be complete only if it (a) fully characterizes the epistemic uncertainties in the model's representation of aleatory variability and (b) can be unconditionally tested (in principle) against observations to identify ontological errors. Unconditional testability, which is the key to model validation, hinges on an *experimental concept* that characterizes hazard events in terms of exchangeable data sequences with well-defined frequencies. We illustrate the application of this unified probabilistic framework by describing experimental concepts for the forecasting of tephra fall from Campi Flegrei. Eventually, this example may serve as a guide for the application of the same probabilistic framework to other natural hazards.

## 1 Introduction

Hazards associated with major eruptions and their consequences are highly uncertain owing to the stochastic behaviors of volcanic systems (aleatory variability) as well as our lack of knowledge about these behaviors (epistemic uncertainty). Because such complexity hampers the deterministic prediction of hazards, our goal is to describe them probabilistically (Sparks, 2003; Marzocchi and Bebbington, 2012; Poland and Anderson, 2020). Here we use the term probabilistic volcanic

hazard analysis (PVHA) to indicate the probabilistic forecast of any volcanological event of interest (eruption occurrence, ash fall loading, arrival of a pyroclastic flow, etc).

PVHA outcome may be generally described by the exceedance probability of a positive random variable $X$ (Martin et al., 2004, Selva et al., 2018; Rougier and Beven, 2013; Sandri et al., 2016; Bear-Crozier et al., 2016)

$$f(x) = \Pr(X > x \mid \boldsymbol{H}), \quad x \in (0, \infty) \tag{1}$$

where $\boldsymbol{H}$ is the PVHA model used, $f(x)$ is called the hazard curve (not to be confused with the hazard function commonly used to describe failure rate; from a statistical point of view $f(x)$ is a survival function). $X$ is the (continuous or discrete) hazard intensity of interest in one specific time interval, for example, the tephra fall loading in one specific site, the dynamic pressure of a pyroclastic flow in one sector of the volcano, or the occurrence of an eruption, etc. Ideally, since we never know the true hazard curve, we may include an additional level of uncertainty considering a set of PVHA models; in this case, we have $\{f_i(x), \pi_i\}$ $(i = 1, \ldots, N)$, where $\pi_i$ is the weight of the $i^{\text{th}}$ PVHA model $H_i$, $f_i(x) \equiv f(x|H_i)$ (Rougier and Beven 2013).

The way in which we use the set of hazard curves $f_i(x)$ to estimate PVHA and the meaning of the weights depend on the probabilistic framework adopted. The importance of establishing the probabilistic framework to calculate PVHA cannot be overstated, because the framework implicitly defines the type of uncertainties, the meaning of probability, and consequently the possibility to validate (at least in principle) the PVHA model. Indeed, the qualitative difference between the objective probabilities of what we can see and count and the subjective probabilities of what we think we know has been recognized for 300 years, separating, for example, the frequentist and Bayesian approaches to probabilistic inference (Hacking 1965). Despite recent attempts to unify these two approaches (e.g., Box 1980; Rubin 1984; Bayarri and Berger 2004; Berger, 2004; Gelman and Shalizi 2013), controversy remains over how aleatory variability can be separated from epistemic uncertainty and whether such a dichotomy is actually useful in hazard analysis. For example, in the Bayesian view adopted by many researchers, the only type of uncertainty is epistemic (e.g., NRC, 1997; Bedford and Cooke, 2001; Lindley, 2000; Jaynes, 2003), whereas in the traditional frequentist framework, the only type of uncertainty is aleatory (Hacking, 1965). Most practitioners recognize that distinguishing between different types of uncertainties can be useful in the interpretation of hazard estimates (e.g., Abrahamson and Bommer 2005; IAEA, 2012; Rougier 2013), but the confusion surrounding the topic is evident in the wide variety of schemes proposed for classifying uncertainties: shallow and deep (Stein and Stein, 2013), intra- and inter-model, external and internal (Rougier and Beven 2013), inter- and intra-model (Selva et al., 2013), value and structural uncertainty (e.g., Solomon et al., 2007), quantified measure of uncertainty and confidence on the validity of a finding (IPCC, 2013), model parameters and initial/boundary conditions, and many others (Reilly et al. 2021). It has been unclear whether these classifications are profound categories that must be reflected in the probabilistic framework or merely convenient, model-based divisions (e.g., NRC, 1997; Rougier and Beven, 2013).

The subject of this paper is to underline the importance of the probabilistic framework in PVHA and its practical implications. This framework must be able to (1) establish a coherent and clear hierarchy of different kinds of uncertainty

(aleatory variability, epistemic uncertainty, and ontological error), (2) assimilate subjective expert judgment into probabilistic models, and (3) unconditionally test complete probabilistic models against data. In the next sections, through a toy example and an application to the Campi Flegrei tephra-fall hazard, we describe the unified probabilistic framework developed by Marzocchi and Jordan (2014, 2017, 2018), which satisfies these requirements. To facilitate the reading and comprehension of the paper, we describe the notation and the new terminology both within the manuscript and in a glossary

at the end of the paper.

## 2 A tutorial example to describe a unified probabilistic framework for PVHA

We consider the case in which $f(x)$ is the annual probability to exceed a specific tephra fall threshold $x$ in one site of specific interest around a volcano (the probability of exceedance, or PoE). The results can be generalized in a straightforward way to other time windows and other types of volcanic threats, as well as to eruption forecasting.

The unified probabilistic framework is rooted in the definition of an experimental concept, which allows us to define a hierarchy of uncertainties. Here we introduce it with a tutorial example, which may be easily generalized to more complex cases. We collect the sequence of annual observations for a particular value of the tephra fall loading $x_0$, which we denote by the binary variable $e_i$. That is, $e_i = 0$ when there is no exceedance in the $i^{th}$ year $(x < x_0)$, and $e_i = 1$ when there is at least one exceedance $(x \geq x_0)$. The experimental concept is the judgment of stochastic exchangeability of the sequence $e_i$, i.e.,

the joint probability distributions is invariant to data ordering when conditioned on a set of explanatory variables (Draper *et al*. 1993). This definition of the experimental concept has implications on the assumptions made about the eruptive process. With the above definition, we are assuming that, at the target volcano, eruptions (or the paroxysmal explosive phases) usually last less than one year (the time window used to define the experimental concept) and the inter-event times between consecutive eruptions (or paroxysmal explosive phases) are conditionally independent and mostly larger than one year. Of

course, if we are interested in volcanoes that behave differently, we have to define another more suitable experimental concept. This example has been chosen because it applies to volcanic systems like Campi Flegrei (the real example of the next section), and it presents similarities with seismic hazard to facilitate the comparison.

A theorem by de Finetti (1974) states that a set of events that is judged to be exchangeable (i.e., the events may come or not from different unknown distributions) can be modeled as identical and independently distributed random variables with a

well-defined frequency of occurrence, $\hat{\phi}$. The frequentist interpretation applied to a set of exchangeable events and the use of the Bayesian mathematical apparatus to handle uncertainties of different kind is the reason for use the term "unified" to characterize the probabilistic framework described in this paper.

In our example, $\hat{\phi} = \hat{f}(x_0)$, where $\hat{f}(\cdot)$ is the true hazard curve, and the unknown frequency $\hat{\phi}$ of the exchangeable sequence $e_i$ is the aleatory variability. The estimation of the unknown true aleatory variability is the target of PVHA, and its estimation

is often possible with different models, $\{\phi_i, \pi_i\}$, where $\pi_i$ is the weight of the $i^{th}$ PVHA model $f_i(x)$, and $\phi_i = f_i(x_0)$. The existence of such alternative models for the aleatory uncertainty reflects the existence of a known uncertainty on the true frequency of the experimental concept (the aleatory variability), usually referred to as "epistemic uncertainty". The definition of the weight $\pi_i$ has an unavoidably subjective nature. The weight of one model may be related in some ways to the hindcast performance of that model and/or through expert judgments. In the usual application of the Bayesian framework to hazard analysis, the set of models is considered complete and exhaustive, and the weight of a model is the probability to be the one that should be used (Scherbaum and Kuehn, 2011); a similar interpretation is often adopted when using the Logic Trees (Bommer and Scherbaum, 2008). In the unified framework the weight represents a measure of the forecasting skill of a model with respect to the others (see Marzocchi and Jordan, 2017, for a deep description of this important issue). Then, from the set $\{\phi_i, \pi_i\}$ we estimate the PoE distribution $p(\phi)$ (hereafter we use $p(\cdot)$ and $P(\cdot)$ to indicate the probability density function and the cumulative distribution, respectively). This procedure is non-unique, it contains a degree of subjectivity, but it is unavoidable; for example, choosing only one estimation $\phi_i$, or the weighted average assumes that $p(\phi)$ follows a Dirac distribution centered to this value. We call $p(\phi)$ the "Extended Experts' Distribution (EED)", which describes the full PVHA (Marzocchi and Jordan, 2014).

In contrast to the Bayesian framework, for which all models are "wrong" and model validation is pointless (Lindley, 2000), the unified framework allows model validation. Specifically, we can define an *ontological null hypothesis*, which states that the true aleatory representation of future occurrence of natural events—the data generating process—mimics a sample from the EED that describes the model's epistemic uncertainty. According to the ontological null hypothesis, the true unknown frequency $\hat{\phi}$ of the experimental concept cannot be distinguished from a realization of the EED, i.e., $\hat{\phi} \sim p(\phi)$. If the data are inconsistent with the EED, the ontological null hypothesis can be rejected, which identifies the existence of an *ontological error* (Marzocchi and Jordan, 2014). In other words, the "known unknowns" (epistemic uncertainty) do not necessarily completely characterize the uncertainties, presumably due to effects not captured by the EED—"unknown unknowns" associated with ontological errors.

In practice, collecting sufficient data for this kind of model validation is only feasible for specific sites surrounding active volcanoes with a high frequency of eruptions. For a specific site near a high-risk volcano with a low eruptive frequency, the data are usually insufficient for formal ontological testing. In probabilistic seismic hazard analysis (PSHA), the problem is overcome trading time with space, i.e., considering many sites simultaneously for one or more time windows. This approach requires that exceedances recorded at different sites can be considered statistically independent; although this may become attainable in PSHA under some specific considerations, it clearly does not hold for sites surroundings a single volcano. In volcanology, trading time with space is useful in validating models for global PVHA (e.g., Jenkins et al., 2015), which consider the eruptive activity of all volcanoes of a specific type. In this case, it is possible to select sites that are far enough to consider the observed exceedances conditionally independent one from each other, and the exchangeable sequence $e_i$ ($i = 1, \ldots, N$) is given by the annual exceedances observed at different $N$ sites. In summary, although the possibility to validate a

PVHA model is conceptually feasible through the unified probabilistic framework (whereas it is not in the Bayesian framework), it is currently often not practically feasible. In fact, the test of an ontological hypothesis (model validation) requires i) the definition of a proper experimental concept, ii) a complete PVHA including aleatory variability and epistemic uncertainty, and iii) enough independent data for testing. We are not aware of cases in which these three requirements are presently satisfied. But we argue that they could be achievable in future dedicated efforts, for example adopting procedures to standardize variables and group "analogue" and exchangeable volcanoes to increase the available data for testing (Tierz et al., 2019).

Future applications may also take advantage from the fact that the exchangeability judgment can be generalized beyond the stationarity of the process (implicit in our example) to more complex situations. For example, we may distinguish ash fall exceedances in the winter and summer seasons, because ash fall loading may be markedly affected by the seasonal dominant winds blowing in different directions. In this case, the data-generating process provides two sequences, $\{e_i^{(1)}: i = 1, \ldots, N_1\}$ for the $N_1$ winter seasons, and $\{e_i^{(0)}: i = 1, \ldots, N_0\}$ for the $N_0$ summer seasons. Both are judged to be Bernoulli sequences and they are observed to sum to $k_0$ and $k_1$ respectively. If the site is located downwind in the winter season, then the expected frequency of $\hat{\phi}^{(1)} = k_1/N_1$ might be greater than that of $\hat{\phi}^{(0)} = k_0/N_0$. As this example makes clear, it is neither the aleatory variability intrinsic to the model that matters in testing, nor the undisciplined randomness of the physical world, but rather the aleatory variability is defined by the exchangeability judgments of the experimental concept. In other words, aleatory variability is an observable behavior of the data-generating process conditioned by the experimental concept to have well-defined frequencies (Marzocchi and Jordan, 2014).

## 3 Accounting for epistemic uncertainty and aleatory variability: the unified framework applied to the tephra fall PVHA at Campi Flegrei

In this section we apply the probabilistic framework outlined in section 2 to the tephra fall PVHA at Campi Flegrei. Although the low eruption frequency of this volcano makes model validation unrealistic in the human time frame, the probabilistic framework has the advantage of providing a full description of the PVHA, accounting for all uncertainties; this may be of particular importance for decision-makers because, for example, they can immediately appreciate the level of uncertainty over the probabilistic assessment made by volcanologists.

Most (if not all) of the studies available for tephra fall PVHA at Campi Flegrei are based on Event Trees (ET; see Newhall and Hoblitt, 2002; Marzocchi et al., 2004; 2008; 2010; Marti et al., 2008; Sobradelo and Marti, 2010). The ET is a popular tree graph representation of events in which individual branches at each node point to different possible events, states, or conditions through increasingly specific subsequent events (intermediate outcomes) to final outcomes; in this way, an ET shows all relevant possible outcomes of volcanic unrest at progressively higher degrees of detail. The probability of each outcome is calculated combining the conditional probability of each branch belonging to the path from the first node to the

final outcome through classical probability theorems. By construction, ET is meant to describe only the intrinsic variability of the process (aleatory variability) and not the epistemic uncertainty, hence it may produce only one single probabilistic assessment $f(x)$. The ET has been generalized to account for uncertainties of different kind replacing the probability at each node with a distribution of probability (Bayesian Even Tree, BET; Marzocchi et al., 2004; 2008; 2010; Neri et al., 2008; Sobradelo and Marti, 2010), which aims at accounting for experts' judgment, different conceptual or physical models, data from analog volcanoes, as well as data from the target or analog volcanoes (Marzocchi et al., 2004; Tierz et al., 2019).

The BET approach fits quite well the unified probabilistic framework that we advocate in this paper. If we consider an experimental concept given by an exchangeable sequence $e_i$ of annual tephra fall exceedances, the outcome of the BET code may be seen as an EED, i.e., the distribution of probability $p(\phi)$ mimics where the true unknown frequency of the exchangeable sequence may be. In the first paper that describes BET (Marzocchi et al., 2004), the use of a distribution of probability at each node was generically advocated to account for the aleatory variability and epistemic uncertainty, using the loose definition of irreducible randomness and limited knowledge of the process, respectively. This paper provides a formal probabilistic background to justify the BET feature of considering the probability at each node as a distribution ($p(\phi)$) instead of a single number; at each node of the event tree, the central value of the distribution is the best guess of the (unknown) long-term frequency of the experimental concept for that specific node (aleatory variability), and the dispersion mimics the uncertainty over this unknown frequency.

The BET approach has been widely investigated for tephra fall PVHA at Campi Flegrei, adopting different choices, hypotheses, and models (Selva et al., 2010; 2018; Sandri et al., 2016). For instance, Selva et al (2018) show the outcomes of five different BET configurations (Figure 1) for one specific site inside the caldera (Figure 2), which differ in the implementation of the tephra fall dispersion model (aggregation and granulometry). In Figure 3 we show the PoE distribution $p(\phi)$ of each configuration for the tephra loading threshold of 300 kg/m$^2$. Each one of the five EED distributions allows for a formal testing of the ontological hypothesis. Note that in this specific case, the EEDs do not account for seasonal variabilities and are relative to the conditional PVHA (conditioned to the occurrence of an eruption). The generalization to an unconditional PVHA can be straightforwardly made, for example, convolving all these distributions with the distribution of the annual eruption probability.

When all EEDs are significantly overlapping (as for many points inside the Campi Flegrei caldera), it means that each BET configuration describes the epistemic uncertainty in a consistent manner. Instead, for the site in Figure 2, we infer that inconsistent BET outcomes (Figure 3) may be due to an underestimation of the epistemic uncertainty in each EED. For this reason, in the example of Figure 3 we consider only the weighted average of each EEDs and then we build a new EED which describes more satisfactorily the overall epistemic uncertainty given by the five BET configurations. This is equivalent to the case of using alternative implementations of the classical ET (Newhall and Pallister, 2015), which produces a set of hazard curves like in the upper panels of Figure 1. In the specific case of Figure 3, the reason for which the epistemic uncertainty is

underestimated in each EED may be due to the BET set up and/or to limitations of BET model to handle some sources of epistemic uncertainty.

The way in which we can build a single EED from a set of point forecasts $\{\phi_l, \pi_l\}$ (lower panels of Figure 1), or from a set of inconsistent EEDs (Figure 3), is what we call *ensemble modeling*. The terms ensemble modeling and ensemble forecasts have been used in many disciplines in different ways since early seventies (e.g., Leith, 1974). The book by Nate Silver (Silver, 2012) gives a wide range of successful applications and uses of ensemble modeling. The common feature across all these different flavors of ensemble modeling/forecasting is the attempt to account for the aleatory variability and the epistemic uncertainty by merging the forecasts of different models or parametrization of the same model in a proper way.

Standard methods are available for the induction of the EED $p(\phi)$ from the set $\{\phi_i, \pi_i\}$. Regardless the details of the procedure, we underline that the nonunique extrapolation of $\{\phi_i, \pi_i\}$ onto the continuous distribution $p(\phi)$ can contribute to ontological errors in the EED. Nonetheless, any reasonable procedure is not more subjective, and certainly less critical, than either not considering the epistemic uncertainty (for example, when using the weighted average $\bar{\phi}$), or assuming that the set of forecasts explore completely and exhaustively the epistemic uncertainty. In the first case, it is assumed that $p(\phi) \equiv \delta(\Phi - \bar{\phi})$ is the Dirac distribution; in the latter case, it is assumed that the proper distribution for the epistemic uncertainty is fully described by the set of forecasts (Marzocchi et al., 2015).

Although ensemble modeling does not prescribe any specific procedure to estimate $p(\phi)$, in statistics random variables bounded in the range [0,1] are often modeled by means of the Beta distribution (Gelman et al., 2003). Hence, we assume that $\Phi \sim \text{Beta}(\alpha, \beta)$, where the parameters $\alpha$ and $\beta$ are related to the weighted average $E(\phi)$ and variance $\text{var}(\phi)$ of $\{\phi_i, \pi_i\}$ through:

$$E(\phi) = \frac{\alpha}{(\alpha + \beta)} \tag{2}$$

and

$$\text{var}(\phi) = \frac{\alpha\beta}{(\alpha + \beta)^2(\alpha + \beta + 1)} \tag{3}$$

Inverting equations 2 and 3, we can get the parameters of the Beta distribution $p(\phi)$, which describes the ontological hypothesis. Using the weights reported in Figure 1, we get the Beta distribution reported in Figure 4. As expected, this global EED is wider than each single EED of the BET configurations reported in Figure 3, and it aims at describing more realistically the complete uncertainty in PVHA. Although the choice of the Beta distribution is subjective, we advocate the use of a unimodal distribution (the Beta distribution is almost always unimodal), which describes more realistically, in most cases, the epistemic uncertainty over the true (unknown) frequency. (See the discussion of Figure 6 in Marzocchi et al., 2015.)

# 4 Discussion and conclusions

In this paper we have described a unified probabilistic framework which allows volcanologists to provide a complete description of PVHA, to define a clear taxonomy of uncertainties (aleatory variability, epistemic uncertainty, and ontological errors), and to account for experts' judgments preserving the possibility to unconditionally test PVHA against data, at least for high-frequency erupting volcanoes, or for global forecasting models. Although in this paper we focus entirely on PVHA, we think that this approach may potentially inspire other scientists working on different natural hazards.

One remarkable and distinctive feature of this probabilistic framework is that the mathematical description of PVHA is given by a distribution of probability (see, e.g., Marzocchi et al., 2004; 2008; 2010; Neri et al., 2008; Sobradelo and Marti, 2010; Bevilacqua et al., 2015), or, equivalently, through a bunch of hazard curves $f_i(x)$ $(i = 1, \ldots, N)$ (Rougier and Beven, 2013). The use of a distribution of probability instead of single numbers mark the main difference with probabilistic frameworks that are more commonly used in PVHA (Marzocchi and Bebbington, 2012), i.e., the frequentist (e.g., Bebbington, 2010; Deligne et al., 2010), and subjective Bayesian (Aspinall et al., 2003). Although these probabilistic frameworks are both legitimate because they are coherent with the Kolmogorov's axioms, they cannot provide a complete description of PVHA, because they cannot unambiguously distinguish and handle properly uncertainties of different kind, which are likely pervasive in natural systems (Marzocchi and Jordan, 2017; 2018).

Making explicit the probabilistic framework in PVHA is important. In the past, loose definitions of the probabilistic framework have provoked critiques of natural hazard analysis (see, e.g., Castanos and Lomnitz, 2002; Mulargia et al., 2017). For example, a vague definition of the nature of uncertainties and the role of subjective judgments brought some scientists to assert that (Stark, 2017) "what appears to be impressive 'science' is in fact an artificial amplification of the opinions and ad hoc choices built into the model, which has a heuristic basis rather than a tested (or even testable) scientific basis." This criticism is implicitly rooted in the (false) syllogism: science is objective, natural hazard analysis relies on subjective experts' judgment, hence natural hazard analysis is not science.

The unified probabilistic framework proposed here emphasizes the importance of model validation (at least in principle) as cornerstone of science; pure objectivity is a myth even in science and the presence of unavoidable subjectivity in PVHA cannot be used to dismiss its scientific nature. In an extreme case, experts' can behave like "models" expressing their subjective measure of the frequency of one defined experimental concept; mutatis mutandis, the same applies to the famous case of farmers who subjectively guess the weight of an ox (Galton, 1907), whose similarities and differences with natural hazard analysis has been discussed in Marzocchi and Jordan (2014). Conversely, the Bayesian framework, which is a full legitimate probabilistic framework to be used in PVHA, does not allow model validation; in this framework all models are wrong (hence, why waste time to validate them?), and we can only evaluate the relative forecasting performance of one model against the others (Lindley, 2000; Jaynes, 2003).

Besides the scientific aspects, the use of a PoE distribution has remarkable practical merits, because it shows to the decision makers both our best guess and the associated uncertainty. In plain words, if two PVHA have the same average, but with quite different variance, this may affect significantly the way in which PVHA could be used by decision-makers. For example, let us consider a case in which there is a critical threshold in PVHA that triggers a specific mitigation action when overcome (this is just a simplified example, because the decision-making has to be based on risk, not on hazard); both averages may be lower of such a critical threshold (hence, both suggesting no action), but, when considering the variance, one of the EED shows a significant part of the distribution above the threshold (suggesting to take action). In this case, the decision-makers may take into consideration the epistemic uncertainty deciding, for the sake of precautionary reasons, to use one specific high percentile of the EED, instead of the average; for example, the Ministry of Civil Defence & Emergency Management in New Zealand (MCDEM, 2008) uses 84[th] percentile of the tsunami hazard analysis as a threshold for taking actions.

As a final consideration, owing to the social implications, we think that only adopting a clear probabilistic framework to get a complete PVHA is the best way to defend probabilistic assessments against future scrutiny and criticism and to use these assessments in the most profitable way.

**Glossary and notation**

This glossary contains the statistical notation used in this paper, and the definitions of new and uncommon terms.

| | |
|---|---|
| *Aleatory variability* | Intrinsic randomness of the data-generating process. In the uncertainty hierarchy, it is the event frequency defined by the experimental concept. |
| *Epistemic uncertainty* | Lack of knowledge about the data-generating process. In the *uncertainty hierarchy*, it is the modeling uncertainty in the event frequency, where the latter is defined by the *experimental concept*. |
| *Exchangeablility* | A property that the joint probability distribution of a data set is invariant with respect to permutations in the data ordering. |
| *Experimental concept* | Collections of data, observed and not yet observed, that are judged to be exchangeable when conditioned on a set of explanatory variables. The experimental concept defines the uncertainty hierarchy. |
| *Extended experts' distribution (EED)* | The continuous probability distribution sampled by the discrete experts' distribution, used to set up the ontological hypothesis. |

| | |
|---|---|
| *Ontological error* | In the uncertainty hierarchy, an error in a model's quantification of the aleatory variability and epistemic uncertainty, where the dichotomy is defined by the experimental concept. |
| *Ontological hypothesis* | A statistical null hypothesis that the event frequency defined by the experimental concept is a sample from the extended experts' distribution. The rejection of the ontological hypothesis exposes an ontological error. |
| *PoE* | Probability of exceedance, i.e., the probability that one specific parameter pf interest will be overcome in one specific time interval. |
| *PVHA* | Probabilistic volcanic hazard analysis; in this case it describes a forecast of the volcanic activity for any forecasting time window, and for different kind of events (eruption, magnitude, ash fall loading threshold, etc). |
| *Uncertainty hierarchy* | Levels of uncertainty, from aleatory variability to epistemic uncertainty to ontologic error, defined by the experimental concept. |
| $f(x)$ | Hazard function or hazard curve; mathematically, it is a survivor function, as described by equation (1). |
| $\pi_i$ | Weight of the $i^{th}$ model. |
| Symbol $^\wedge$ | It indicates the true value of an unknown (e.g., $\hat{\phi}$ is the true frequency of the experimental concept), or the true probability distribution (e.g., $\hat{f}(\cdot)$ is the true hazard curve). |
| $z \sim p(z)$ | The random variable $z$ is distributed according to the distribution $p(z)$. |
| $p(\cdot)$ and $P(\cdot)$ | Probability density function and the cumulative distribution, respectively |
| $E[z]$ and $\text{var}(z)$ | Expectation and variance of the random variable $z$, respectively |

**Author contribution:** All authors contributed, at different levels, to the conceptualization of the research. WM and JS carried out the formal analysis and numerical investigation. WM wrote the first draft of the paper with contributions from all
270 the other authors. All authors reviewed and edited the final version.

**Acknowledgments:** TO BE WRITTEN [include: THJ was supported by a grant from the W.M. Keck Foundation.]

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

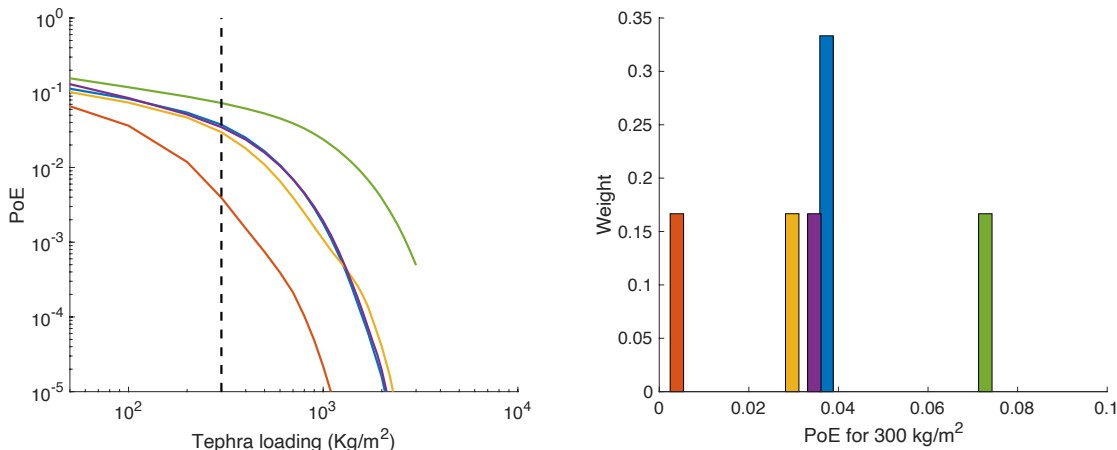

**Figure 1: The left panel shows the mean tephra fall hazard curves relative to one point inside the Campi Flegrei (Figure 2) for each BET implementation described in Selva et al. (2018); colors indicate each implementation. The right panel shows the PoE relative to the tephra load threshold of 300 kg/m² (vertical dashed line in the upper panel), and the weight of each assessment.**

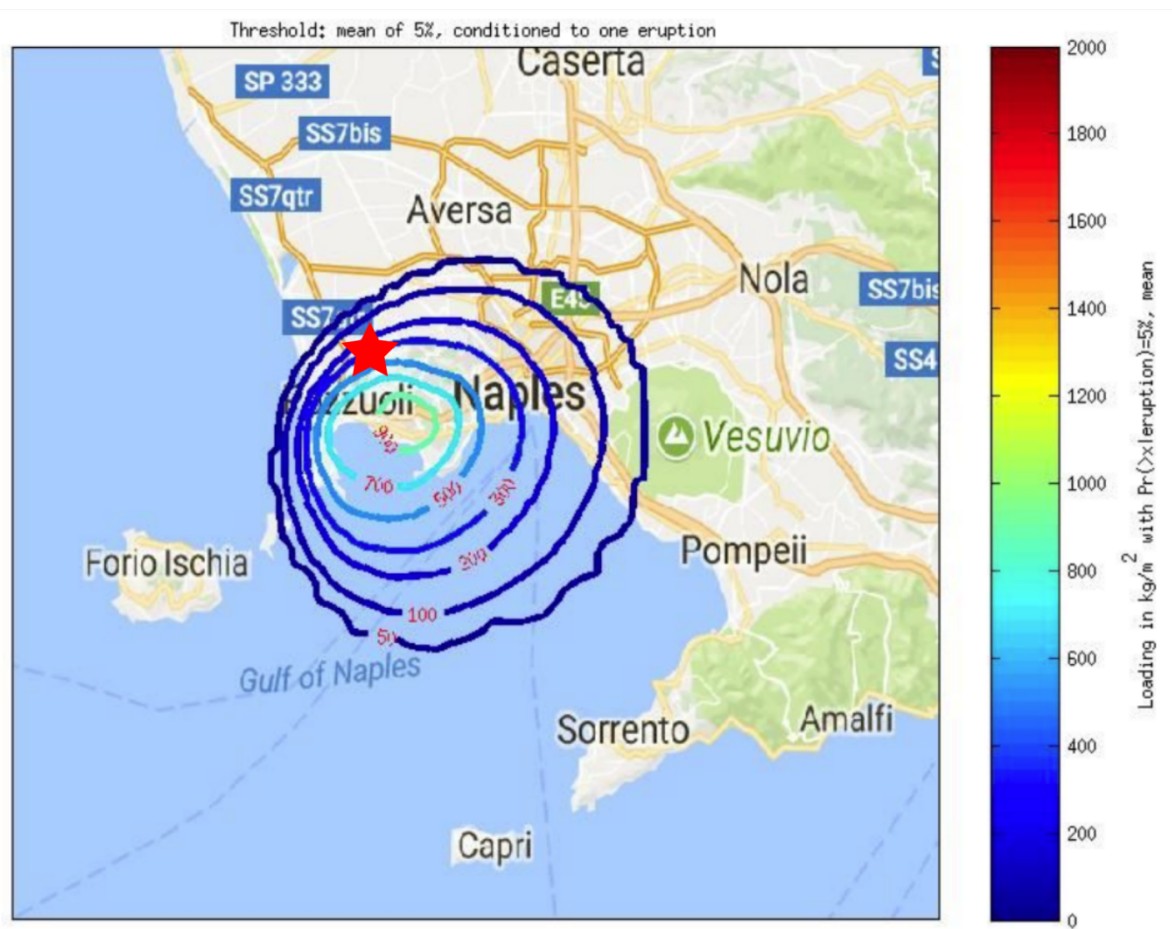

**Figure 2: The location reference site for this study (red star) along with the hazard map relative the probability of 5% of exceedance, conditional upon the occurrence of one eruption of whatever size and from whatever vent at Campi Flegrei (mean of the epistemic uncertainty). The figure has been obtained modifying the Figure 11 of the corrigendum to the paper Selva et al., 2018**

**(available at doi.org/10.1016/j.jvolgeores.2018.07.008); the original maps have been obtained with the MATLAB software.**

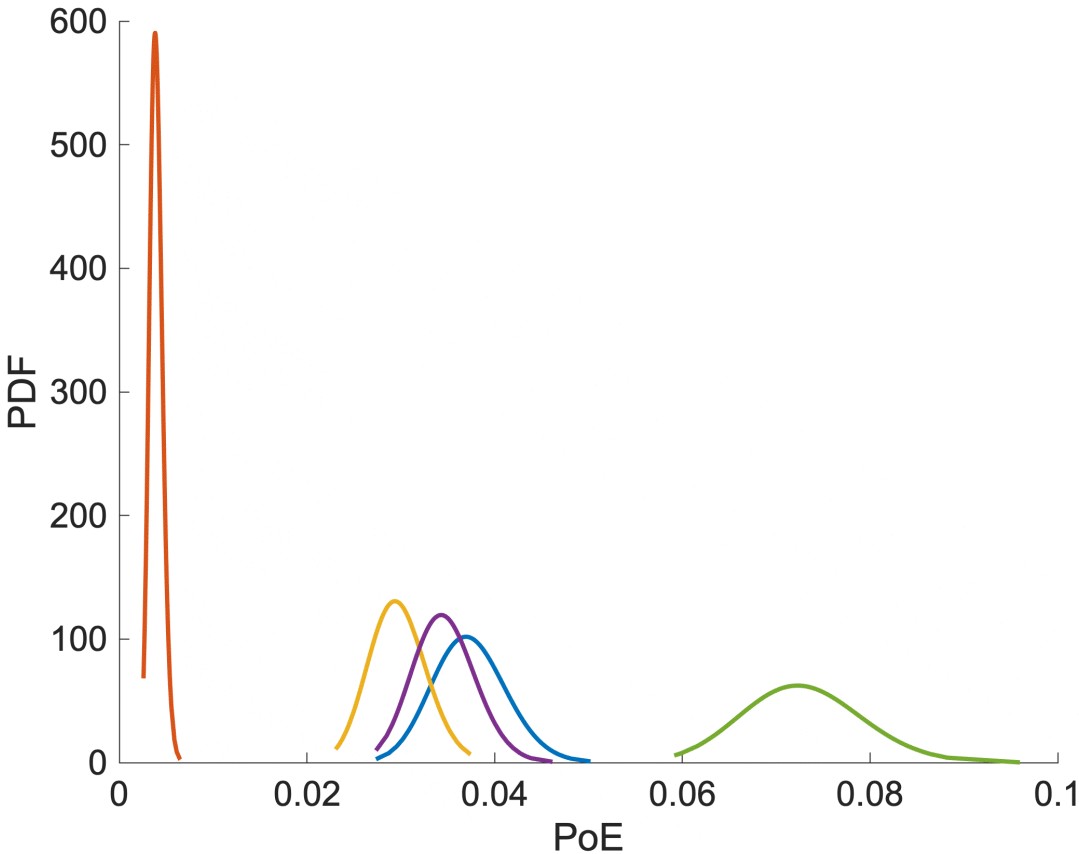

**Figure 3: EED of each BET configuration relative to a tephra load threshold of 300 kg/m² for the site in Figure 2.**

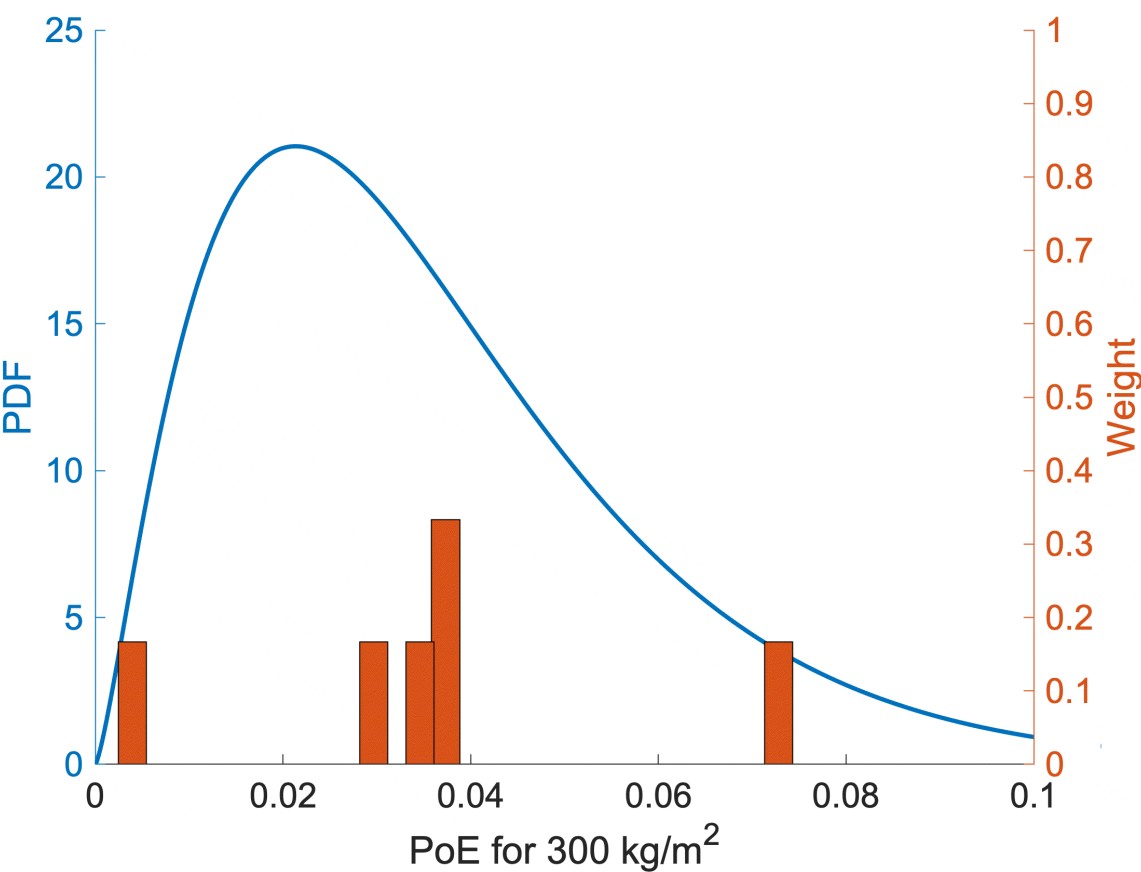

**Figure 4: The overall EED obtained from the average of the five distributions of Figure 3. The EED is a Beta distribution with parameters given by equations 2 and 3. The histogram (right y-axis) shows the weight of each one of the five distributions.**
