# Peer review of "A Unified Probabilistic Framework for Volcanic Hazard and Eruption Forecasting"

_Natural Hazards and Earth System Sciences, 2021_

## Author Response (AR1)

**1 Reviewer 1**

1. As a statistician, I had a lot of problems with the decidedly non-Statistics system of notation. In Statistics, the 'hat' notation is reserved for estimates. In the m/s, the hat notation is used to denote the (unknown) true value. Less important is the use of f() to denote a survival function (in Statistics, \bar{F}() or S() would be used). F would typically be the (unknown) distribution from which the variable is drawn, a niche which in the m/s is occupied by \hat{\phi}. The estimate would be \hat{F}, rather than P(\phi). As the paper is written from a subjectivist viewpoint, should appropriate Bayesian notation be used throughout? I do understand the need for backward compatibility, but at minimum a glossary could be provided.

Answer: We agree that notation is essential. As the reviewer said, we have decided to use this notation to maintain a compatibility with past publications. As suggested by the reviewer, we have added a table containing the description of all symbols and terms that we have used in the manuscript. We agree that it is a good tool to facilitate the reading and comprehension also for pure statisticians that are used a different notation.

2. More clarification could be devoted to the (sometimes fine) distinction between epistemic uncertainty and ontological error. Is the latter simply an extreme case of the former? At Line 89-96, it appears that ontological error is only identified as a consequence of a Bayesian model checking procedure (P-value), and so itself is subject to uncertainty.

Answer: We have added more explanation in the paper. In essence, using a very popular terminology, we may say that epistemic uncertainty is the "known unknowns", whereas the ontological errors are the "unknown unknowns" that, by definition, cannot be included into the models because we are completely ignorant about them, but evidence for them may emerge only in the testing phase.

3. The tutorial example, while mathematically correct, does not seem to reflect an actual problem in volcanology. In practice the actual variable would be exceedance given an eruption, and so i should index the eruption number, to be consistent with the example in Section 3, not the year. Otherwise, as eruptions are not point events in time, exchangeability would be invalidated by whether an eruption was in progress at year begin/end. Presumably the exceedance is measured at a single location, such as a critical installation. Further discussion is needed on the degree to which the magnitude of individual eruptions are exchangeable. Seasonal wind patterns could also be mentioned here for the tephra example.

Answer: We have modified the text explaining that the example reflects the unconditional ash fall hazard in one specific site, such as, for example, a critical infrastructure. In this case, the experimental concept is composed by the exceedances observed (or not) in non-overlapping time windows (1 year) that we consider exchangeable. We do think that this has to be of great interest (and it is) for volcanology, exactly as it is in seismology. In the revised manuscript we have explained the link between the tutorial and real example, that is just linked to multiplying the conditional PVHA of the real example by the probability of
eruption occurrence to obtain the full PVHA of the tutorial example. Additionally,
we have clarified the volcanological assumptions that stand behind the tutorial
example: in particular, the experimental concept adopted assumes that, at the
target volcano, eruptions usually last less than one year and are dominated by one
major ash emission (we have one or no exceedance at the site), and the inter-event
times between consecutive peaks of eruptive activity are conditionally
independent and mostly larger than one year. In other words, more than 1
exceedance per year is an unlikely event, at least for the selected range of tephra
fall loading. Of course, if we are interested in volcanoes that behave differently,
we have to define another more suitable experimental concept. This example has
been chosen because it applies reasonably well to volcanic systems like Campi
Flegrei, and it allows a comparison with seismic hazard.

As regards the problem of the seasonal winds, we have modified the tutorial
example to account for that (see answer to comment 6).

4.  Line 83 states that "The unknown true aleatory variability is often estimated
by different models …", but this seems to be the procedure followed later in
the m/s to estimate the epistemic uncertainty?

Answer: That's correct; this statement is misleading. We have modified this point
to make it clearer.

5.  Need discussion about where $\pi_i$ comes from at Lines 83-84. The notation
in Marzocchi and Jordan (2017) is clearer in this regard.

Answer: We have modified the text accordingly. We have avoided the duplication
of the discussion that we have already reported in Marzocchi and Jordan (2017),
but in the revised version we have mentioned the different kind of weights in
different probabilistic frameworks. In particular, the definition of the weight has
an unavoidably subjective nature. The weight of one model may be related in
some ways to the hindcast performance of that model and/or through expert
judgments. In the Bayesian framework the set of models has to be complete and
exhaustive, and the weight of a model is the probability to be the one that should
be used; a similar interpretation is often adopted when using the Logic Trees. In
the unified framework the weight represents a measure of the forecasting skill of a
model with respect to the others (see then discussion in Marzocchi and Jordan).

6.  At Lines 108-113, the discretization of time is causing further confusion. A
clear distinction would need to be made between an earthquake _preceding_
an eruption and one following it.

Answer: That's correct. Thank you. We have replaced the earthquake example
with one that conditions the ash-fall hazards on the seasonal winds. In particular,
assuming that the dominant winds are different in winter and summer, we can
conceive two different experimental concepts that may be characterized by two
different aleatory variability (different hazard). In this case, the two experimental
concepts are relative to the ash fall exceedances observed in the winter and
summer seasons.

7. I think what the authors are saying in Lines 113-117 is that uncertainty can be apportioned between aleatory and epistemic, and that uncertainty assigned to the former cannot result in ontological error? Some clarification would be welcomed.

Answer: The point of this example (we modified the text accordingly) is just to say that the (true) aleatory variability is not related to the true process governing the Earth, but exclusively to the data-generating process which is related to the experimental concept that we define. If we change the experimental concept, we may change (but not necessarily) the aleatory variability.

8. I don't understand L143-144 in view of the (blurry-)definition of aleatory and epistemic uncertainty earlier in the paper, belying Objective (1) at Lines 64-5. The concepts do not seem to be clearly and consistently separated. From a subjectivist viewpoint, the aleatory uncertainty is a probability distribution, the epistemic uncertainty is a prior on the parameters of the probability distribution, and ontological error is a probability that the aleatory/epistemic system fails to represent the data.

Answer: We have removed that statement because it may be misleading. The point here was to state that the old definition of aleatory variability and epistemic uncertainty is quite blur and it does not allow a clear distinction between different kind of uncertainties.

9. Should "underestimation" at Line 157 be "misestimation"?

Answer: Actually, we consider it as an "underestimation", since the EEDs are narrow and not overlapping. In practice, a narrow distribution means that the model is underestimating the epistemic uncertainty if other legitime models say something completely different.

10. The sentence ending on Line 223 might be overstated. Decision makers have enough difficulty with means, variances may be completely beyond them. There is a considerable body of research on this….

Answer: What the reviewer said is patently right! Currently, most decision makers are struggling to handle the probabilities described by single numbers. However, we do think that the uncertainty over the scientific assessment has to be considered and not disclosed because we think that others cannot understand that. For example, also the IPCC introduced in AR5 a qualitative measure of the epistemic uncertainty through the term "likelihood" and "confidence". The likelihood is the outcome of a model or an ensemble of models (one distribution), and the confidence may be low, medium, or high, equivalent to, in our framework, the epistemic uncertainty given by the dispersion of the EED around the mean.

In essence, it is not the decision maker that can impose what we know and what we do not know, or how the Nature behaves. The decision-makers have to become aware of what we can say, and of the uncertainty that we have, if they want to use our assessment in a rational way. In the revised manuscript we have made a simple example to illustrate this point: let us consider a case in which there is a

| 130 | critical threshold in PVHA that triggers a specific mitigation action when |
| 131 | overcome (this is just a simplified example, because the decision-making has to be |
| 132 | based on risk, not on hazard). We have two different assessments with the same |
| 133 | average and completely different variances. The common average may be lower |
| 134 | than the critical threshold (hence, suggesting no action), but, when considering the |
| 135 | variance, one of the EED shows a significant part of the distribution above the |
| 136 | threshold (suggesting to take action). In this case, the decision-makers may take |
| 137 | into consideration the epistemic uncertainty deciding, for precautionary reasons, |
| 138 | to use one specific high percentile of the EED, instead of the average; for |
| 139 | example, the National Emergency Management Agency in New Zealand |
| 140 | (MCDEM, 2008) uses 84th percentile of the tsunami hazard analysis as a |
| 141 | threshold for taking actions. |

11. As the earlier (seismic) papers refer to the SSHAC guidelines, should similar
(eg. IAEA SSG-21?) citations be made here?

Answer: Done.

Technical corrections

Line 100 "…simultaneously for one …"

Line 166 "… recent book by Nate Silver (Silver, 2012) …"

Line 203 Bebbington (2010) is not in the reference list

Line 232/3 These references are not cited in the text.

Answer: Done

**Reviewer 2**

It is very good to see this exposition of the probabilistic framework for
PVHA.  However, the testing of this framework is disappointing, because of the low
eruption frequency at Campi Flegrei, which is a limitation recognized by the
authors.  The ideal laboratory for testing alternative PVHA methodologies is a
volcano which has sporadic bouts of activity over a decade or more.  An example is
Montserrat from 1995 onwards.  Some attempts have been made to validate
probabilistic forecasts for Montserrat against actual eruptive events, but this has not
been done in a systematic manner, because these were early days in PVHA, and the
resources were limited for updating PVHA regularly.

The paper makes much of the experimental concept of testing model validation, so
there should be a convincing example of such validation.  The convenience for the
authors of Campi Flegrei is of course well appreciated.  However, the authors should
identify a more active laboratory for adequately testing their PVHA approach.

Answer: We thank the reviewer for appreciating the discussion on the
probabilistic framework. However, we do not agree with the fact that Campi

Flegrei is a less interesting example than Montserrat. In essence, at Campi Flegrei we have a complete PVHA made with different models (Figure 1). This allows us to discuss a coherent way to handle the uncertainties, defining an unambiguous hierarchy of uncertainties. This case can be reproduced easily for many volcanoes with a limited effort.

Hence, the problem of testing is of course very important, but it is not the only reason to consider this probabilistic framework. Similar discussions have been made also in long-term seismic hazard; although the validation of the model is practically very unlikely (due to the long time to get several 50 years time windows of data) there has been a long discussion on how to interpret the outcomes of the logic tree, which is a very popular tool to estimate the epistemic uncertainty (a deeper discussion can be found in Marzocchi et al., 2015; Marzocchi and Jordan, 2017). As regards the Montserrat case, the validation of the model could be hard (but maybe solvable; we were not involved in that experience) for two main reasons: first, it is not clear what the experimental concept is; second, a complete forecast (EEDs), which separate aleatory variability and epistemic uncertainty is not available; third, if the subjective framework has been adopted (this is what we perceived from literature), it does not make sense to validate the model. In the subjective framework we can only compare the performance of one model with respect to other competing models (we discuss this topic in Marzocchi and Jordan, 2014).

To conclude, with this paper we do not aim at ending the discussion about the importance of the probabilistic framework in PVHA, or saying which one has to be used. But we do aim to raise awareness on the importance to use one of the legitimate probabilistic frameworks and to remain coherent with that.